# Live calcium and mitochondrial imaging in the enteric nervous system of Parkinson patients and controls

An-Sofie Desmet[1,2†], Carla Cirillo[1,2†], Jan Tack[2], Wim Vandenberghe[3,4*‡], Pieter Vanden Berghe[1,2*‡]

[1]Laboratory for Enteric NeuroScience, University of Leuven, Leuven, Belgium; [2]Translational Research Center for GastroIntestinal Disorders, University of Leuven, Leuven, Belgium; [3]Laboratory for Parkinson Research, Department of Neurosciences, University of Leuven, Leuven, Belgium; [4]Department of Neurology, University Hospitals Leuven, Leuven, Belgium

**Abstract** Parkinson's disease (PD) is a neurodegenerative disease with motor and non-motor symptoms, including constipation. Therefore, several studies have investigated the gastrointestinal tract, and more specifically the enteric nervous system (ENS), in search of an early biomarker of PD. Besides α-synuclein aggregation, mitochondrial dysfunction and dysregulation of intracellular $Ca^{2+}$ concentration probably contribute to the pathogenesis of PD. Here we assessed neuronal and mitochondrial functioning in primary enteric neurons of PD patients and their healthy partners as controls. Using a unique combination of live microscopy techniques, applied to routine duodenum biopsies, we were able to record neuronal $Ca^{2+}$ responses and mitochondrial membrane potential in these nerve tissues. We found that submucous neurons were not affected in PD patients, which suggests that these neurons are not involved in the pathogenesis or the gastrointestinal symptoms of PD. Our study provides for the first time functional information on live neurons in PD patients.

*For correspondence: wim.
vandenberghe@uzleuven.be (WV);
pieter.vandenberghe@med.
kuleuven.be (PVB)

†These authors contributed
equally to this work
‡These authors also contributed
equally to this work

**Competing interests:** The authors declare that no competing interests exist.

## Introduction

Parkinson's disease (PD) is the most prevalent neurodegenerative movement disorder. The defining pathological features of PD are the loss of dopaminergic neurons in the substantia nigra (SN) and the intraneuronal presence of α-synuclein inclusions (Lewy bodies and Lewy neurites) (*Lees et al., 2009*). Although the pathogenic mechanisms underlying PD are not understood in detail, mitochondrial dysfunction and dysregulation of calcium homeostasis are thought to play a crucial role (*Exner et al., 2012*; *Surmeier et al., 2013*).

Besides the well-known motor problems, PD patients can develop a variety of disabling non-motor symptoms (*Lees et al., 2009*), like psychosis, depression, hyposmia, rapid eye movement (REM) sleep behavior disorder and constipation, some of which can even occur prior to the first motor manifestations (*Goldman and Postuma, 2014*). This has sparked a growing interest in probing non-motor aspects for early diagnosis. Gastro-intestinal (GI) dysfunction in PD has recently attracted a lot of attention in this respect (*Fasano et al., 2015*). Several studies have reported the presence of α-synuclein aggregates in the enteric nervous system (ENS), which controls GI function, in fixed biopsy material or postmortem tissue from PD patients (*Lebouvier et al., 2008*; *Shannon et al., 2012a*; *Pouclet et al., 2012a*; *Derkinderen et al., 2011*) suggesting that the ENS is directly affected by the disease process. This finding also fueled the hypothesis that α-synuclein pathology may spread from the periphery to the brain. According to this theory, an ingested pathogenic agent would enter nerve fibers in the GI tract and initiate α-synuclein misfolding, which would

then propagate in a prion-like fashion along the axons up to the dorsal motor nucleus of the vagus in the lower brainstem (*Braak et al., 2006*; *Hawkes et al., 2007*). Nevertheless, more recent studies have shown similar patterns of α-synuclein immunoreactivity in the ENS in a high percentage of neurologically unimpaired controls (*Visanji et al., 2014*; *Gold et al., 2013*; *Gray et al., 2014*). Given the current debate about the potential utility of enteric α-synuclein immunohistochemistry as a biomarker for PD (*Visanji et al., 2014*; *Ruffmann and Parkkinen, 2016*), new approaches are warranted to measure the involvement of the ENS in PD.

So far, not a single report has investigated the functionality of enteric neurons in PD patients. The general aim of this study was to examine the functionality of living enteric neurons of well-characterized PD patients. We used calcium imaging (*Cirillo et al., 2015*, *2013*) as a reliable proxy to assess neuronal function and mitochondrial imaging, to test the functionality of enteric neurons and mitochondria in freshly isolated submucous plexus preparations from PD patients and controls (*Figure 1*).

## Results

### Study population

We recruited 15 couples, each consisting of a PD patient and his or her healthy partner. This pairwise recruitment allowed within-pair comparisons to better control for variability in diet, lifestyle and other environmental factors. Demographic and clinical characteristics are summarized in *Table 1* and *Supplementary file 1*. As in most clinical PD studies, the majority of PD participants were male. PD patients and controls were age-matched and disease duration ranged from 2 to 17 years. All PD patients were under treatment with oral dopaminergic medication. None were treated with apomorphine, levodopa-carbidopa intestinal gel or deep brain stimulation. Three patients had levodopa-induced dyskinesias, five had early morning dystonia and two had daytime motor fluctuations. None of the patients had a first-degree relative with PD, except possibly one whose mother had allegedly developed tremor around the age of 90 years. Six of the 15 patients had disease onset under the

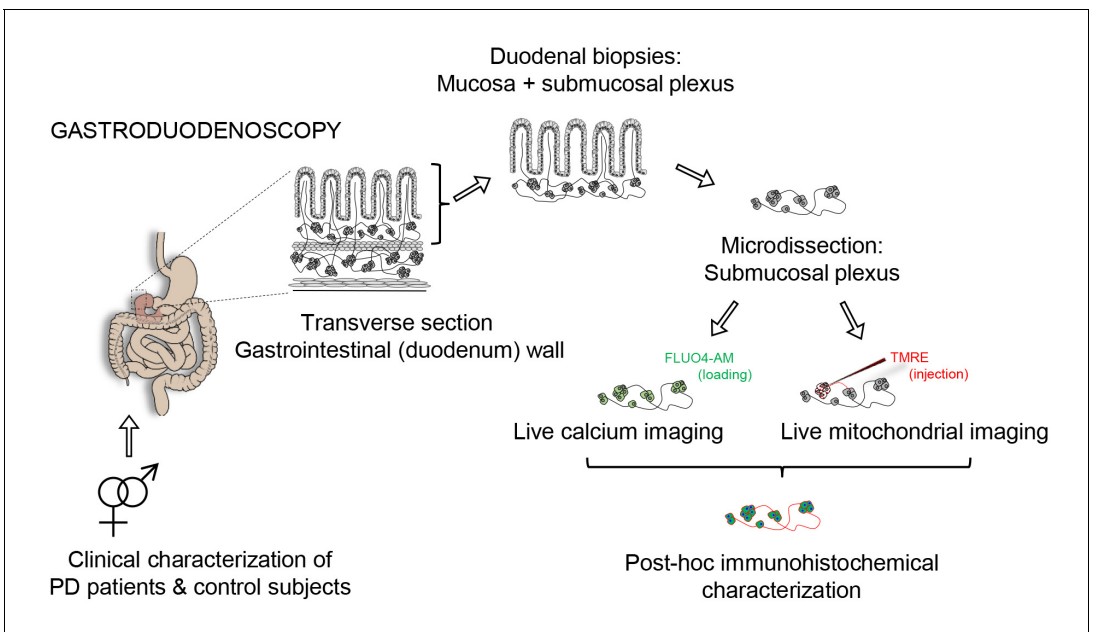

**Figure 1.** Schematic representation of the experimental strategy. Gastroduodenoscopy was performed on clinically well-characterized PD patients and their healthy partners, and 8 biopsies of the duodenum were taken per subject. The submucous plexus was peeled away from the mucosal epithelium and was used for live imaging techniques ($Ca^{2+}$ or mitochondrial imaging) followed by post-hoc immunohistochemistry for confirmation of neuronal identity. In addition, the submucous plexus was isolated from 3 fresh biopsies per subject and immediately processed for immunohistochemistry (without live imaging) and numbers of neurons and ganglia were counted per biopsy (not indicated in the schematic).

**Table 1.** Demographic and clinical characterization of PD patients and controls Mean ± SD are shown with associated p-value (non-parametric Wilcoxon T-test[a] | Wilcoxson T-test [b] | Chi squared test [c]).

| | Control subjects (n = 15) | PD patients (n = 15) | p-value |
|---|---|---|---|
| Gender (M:F) | 4: 11 | 11: 4 | 0.01 (*) [C] |
| Age | 57.8 ± 2.6 (range: 44–76) | 58.9 ± 9.2 (range: 45–71) | 0.40 [a] |
| SCOPA total | 5.6 ± 2.7 (range: 2–10) | 12.3 ± 9.2 (range: 2–32) | 0.03 (*) [a] |
| SCOPA GI | 0.9 ± 0.80 (range: 0–2) | 2.7 ± 2.29 (range: 0–8) | 0.02 (*) [b] |
| Disease duration (years) | - | 7.8 ± 3.9 (range: 2–17) | |
| UPDRS III off (disease severity) | - | 23.3 ± 10.0 (range: 12–46) | |
| Age at onset (years) | - | 51.1 ± 9.4 (range: 36–69) | |
| Hoehn-Yahr (off) | - | 2 (IQR: 2–5) | |
| LED (mg) | - | 684.1 ± 388.5 (range: 205–1740) | |
| MMSE | - | 29.6 ± 0.9 | |

(*indicates a statistical difference p<0.05). For Hoehn-Yahr scores, median and interquartile range are shown.

age of 46 years. Genetic analysis of the *PARK2* gene was performed in 5 of these 6 patients, but no mutations were found.

Interestingly, although presence of autonomic and GI symptoms was not an inclusion criterion, PD patients had significantly more autonomic and GI symptoms than controls (*Table 1*).

## $Ca^{2+}$ signalling properties of submucous neurons do not differ between PD patients and controls

Neurons were identified based on their specific morphology, localization in a ganglion and characteristic Fluo-4 loading (*Figure 2A*, left), as previously described (*Cirillo et al., 2015, 2013*), and neuronal viability was assessed using a short high $K^+$ depolarization (10 s, 75 mM $K^+$) (*Figure 2A*, left) (*Video 1*), which induced a transient rise in intracellular calcium $[Ca^{2+}]_i$ (*Figure 2A*, middle). Post-hoc immunostaining for the neuronal markers HuCD and NF200 confirmed the neuronal identity of the cells selected during the live recordings (*Figure 2A*, right). In controls, 57.3 ± 29.5% of neurons displayed a transient change in Fluo-4 intensity (with maximum amplitude of 3.6 ± 2%, n = 15.6 ± 2.9 neurons per subject) upon depolarization with high $K^+$. No significant difference was found when compared to PD patients, where both the number of high $K^+$ responding neurons (49.5 ± 28.7%, p=0.47) and the $[Ca^{2+}]_i$ transient amplitudes (2.8 ± 3%, p=0.37, n = 11.9 ± 1.9 neurons per patient) were similar (*Figure 2B*). The percentage of responding neurons and $[Ca^{2+}]_i$ transient amplitudes in the PD group did not correlate with age, GI symptoms (as measured by the Scale for Outcomes in Parkinson's disease for Autonomic Symptoms (SCOPA-AUT), disease duration or disease severity (as assessed by Unified Parkinson's Disease Rating Scale (UPDRS) part III off medication (*Supplementary file 2*).

In addition to high $K^+$ depolarization we also used two more physiological stimuli to assess neuronal function: 1,1-dimethyl-4-phenylpiperazinium (DMPP), a nicotinic acetylcholine receptor agonist, as fast excitatory transmission in the ENS occurs mainly via acetylcholine on nicotinic receptors; and trains of electrical pulses applied to the fiber strands in the submucous plexus. However, neither for DMPP nor for electrical stimulation, differences in percentages of responding neurons or response amplitudes were found (*Figure 2B*).

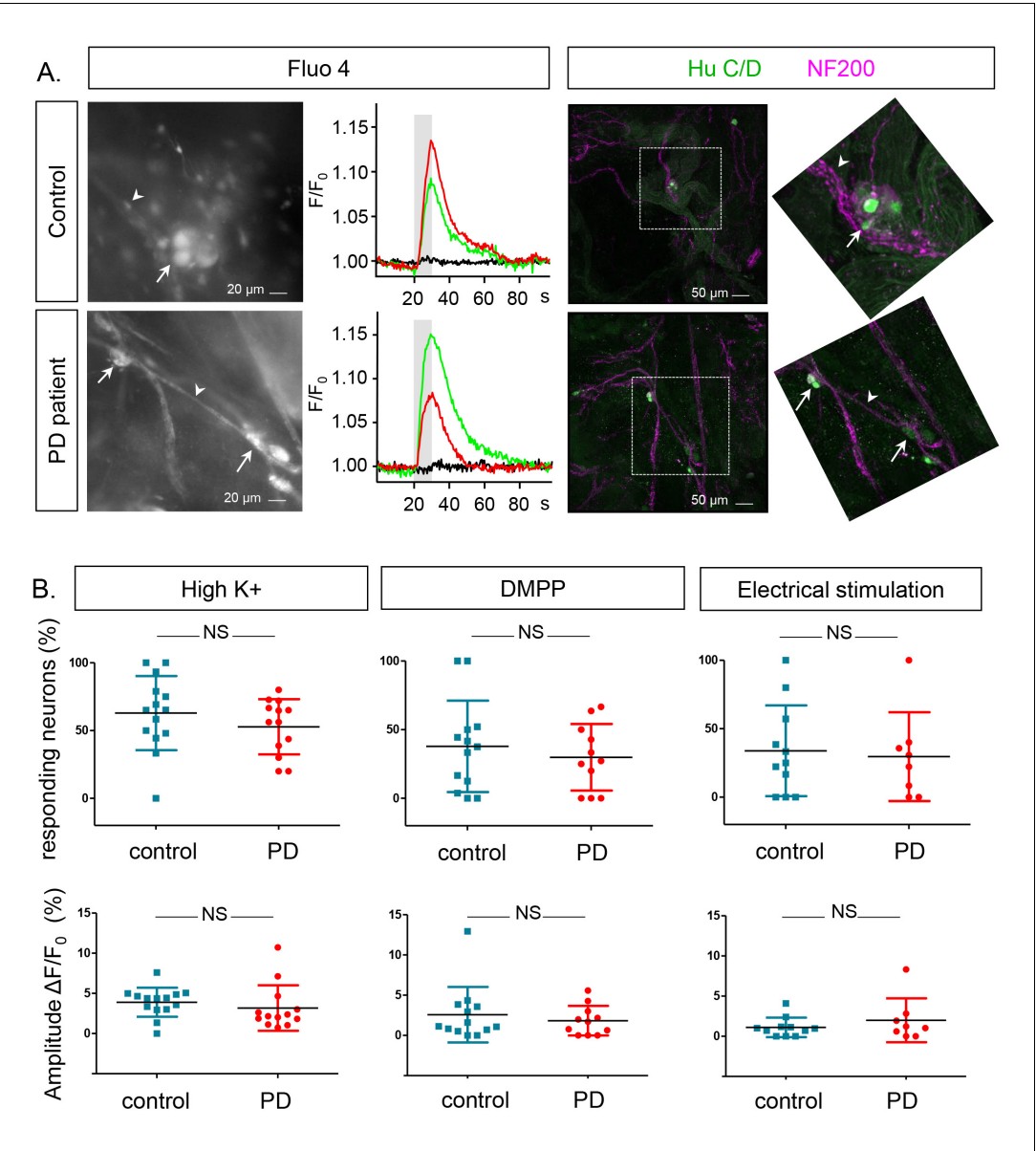

**Figure 2.** $Ca^{2+}$ signalling in submucous neurons in biopsies from PD patients and controls. (**A.**) Representative examples calcium imaging (left) and posthoc immunohistochemical staining (right). A typical example of submucous ganglia (PD and control) loaded with Fluo-4 is shown (left) as well as typical $Ca^{2+}$ traces (normalized to the fluorescence at time zero $F/F_0$) of single submucous neurons depolarized with a high $K^+$ stimulus (represented as grey bar). The green and red traces are 2 random examples, the black trace is the background value. The panels on the right show the correlated post-hoc immunostaining for enteric neuronal markers (Green: HuCD (neuronal cell bodies); Magenta: NF200 (intermediate neurofilament 200: neuronal fibers). The inset shows a magnification of the selected ganglion (dashed square). The arrow (neuronal cell bodies) and arrowheads (neuronal fibers) point to the same structures in both immunohistochemical and corresponding Fluo-4 images (left). (**B.**) Summary data plots showing the percentage of responding neurons (top row) and maximum peak amplitude $\Delta F/F_0$ (%) (bottom row) for high $K^+$, DMPP and electrical stimulation. The individual data points represent a patient or control for which all individual neuronal responses were averaged. (NS, not significantly different, non-parametric Wilcoxon T-test; p-values: % of responders; High $K^+$=0.23, DMPP = 0.88, ES = 0.99 | amplitudes: High $K^+$=0.57, DMPP = 0.99, ES = 0.81).

The following source data is available for figure 2:

**Source data 1.** Calcium imaging % responders.

*Figure 2 continued on next page*

*Figure 2 continued*

**Source data 2.** Calcium imaging amplitude.

## Mitochondrial membrane potentials of submucous neurons are similar in PD patients and controls

To assess mitochondrial membrane potential, submucous ganglia were injected with Tetramethylr-hodamine, ethyl ester (TMRE) and 3D confocal recordings were made and quantified using Andor iQ and IMARIS (*Figure 3A*). First, the average intensity (in arbitrary values) was determined in record-ings made at room temperature (RT). The TMRE signals at RT were similar in both groups (*Figure 3B*) and did not correlate with age, GI symptoms, disease duration or severity of PD patients (*Supplementary file 3*). To monitor mitochondrial membrane potential changes, preparations were kept at 37°C while TMRE signals were measured continuously over several minutes(*Videos 2–3*). The intensity of the TMRE staining fluctuated substantially, indicating that both in PD patients and con-trols mitochondrial potentials were dynamically changing (*Figure 3C–D*). Here again, TMRE intensity fluctuations were not significantly different between PD patients and controls (*Figure 3E*), without any correlations with the clinical characteristics of PD patients (*Supplementary file 3*).

## Mitochondrial volume and numbers in submucous neurons do not differ between PD patients and controls

Given the similarity of TMRE loading between PD patients and controls and the problems with mito-tracker green (MTG) loading (see Materials and methods), we used the TMRE signals recorded at RT to also compare mitochondrial volume and numbers (*Figure 4A*). We first analysed the total mito-chondrial volume and number of mitochondria in a selected 3D mask surrounding the injected gan-glion (*Figure 4A*). The volumes of the outlined ganglia were not significantly different between the 2 groups (data not shown), nor was the total mitochondrial volume inside the selection (*Figure 4B*). The mitochondrial volume did not significantly correlate with age, GI symptoms, disease duration or severity of PD patients (*Supplementary file 3*). In addition to total mitochondrial volume, we also analysed mitochondrial density (number of mito-chondria within the ganglion outline). This param-eter was again not significantly different between the two groups (*Figure 4C*) and did not correlate with clinical characteristics in the PD group (*Supplementary file 3*). Lastly, we compared average volume of individual mitochondria and, similarly to other mitochondrial parameters, this did not significantly differ between the two groups (*Figure 4D*), nor was it correlated with any of the clinical parameters (*Supplementary file 3*) (*Video 4*).

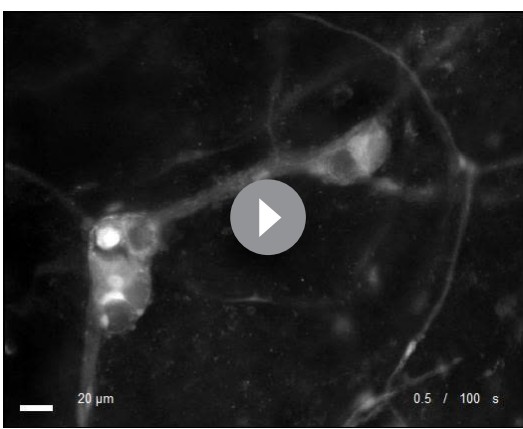

**Video 1.** Calcium imaging in human submucous neurons. This movie shows a representative recording of Fluo-4 intensity changes in 2 human submucous ganglia (the ganglion on the right holds 2, while the one the left contains 5 neurons) during high K$^+$ stimulation (10 s). The movie, which is representative for both groups (patients and controls) was taken from a PD patient sample. The original recording was deconvolved using Huygens software and movies generated using IGOR pro and ImageJ.

## The submucous plexus contains similar numbers of ganglia and neurons in PD patients and controls

Finally, we counted the number of neurons per ganglion, total number of neurons per biopsy and number of ganglia per biopsy in both groups based on immunofluorescent staining (*Figure 5A*). None of these parameters differed between the two groups (*Figure 5B–D*). No cor-relations were found between these counts and

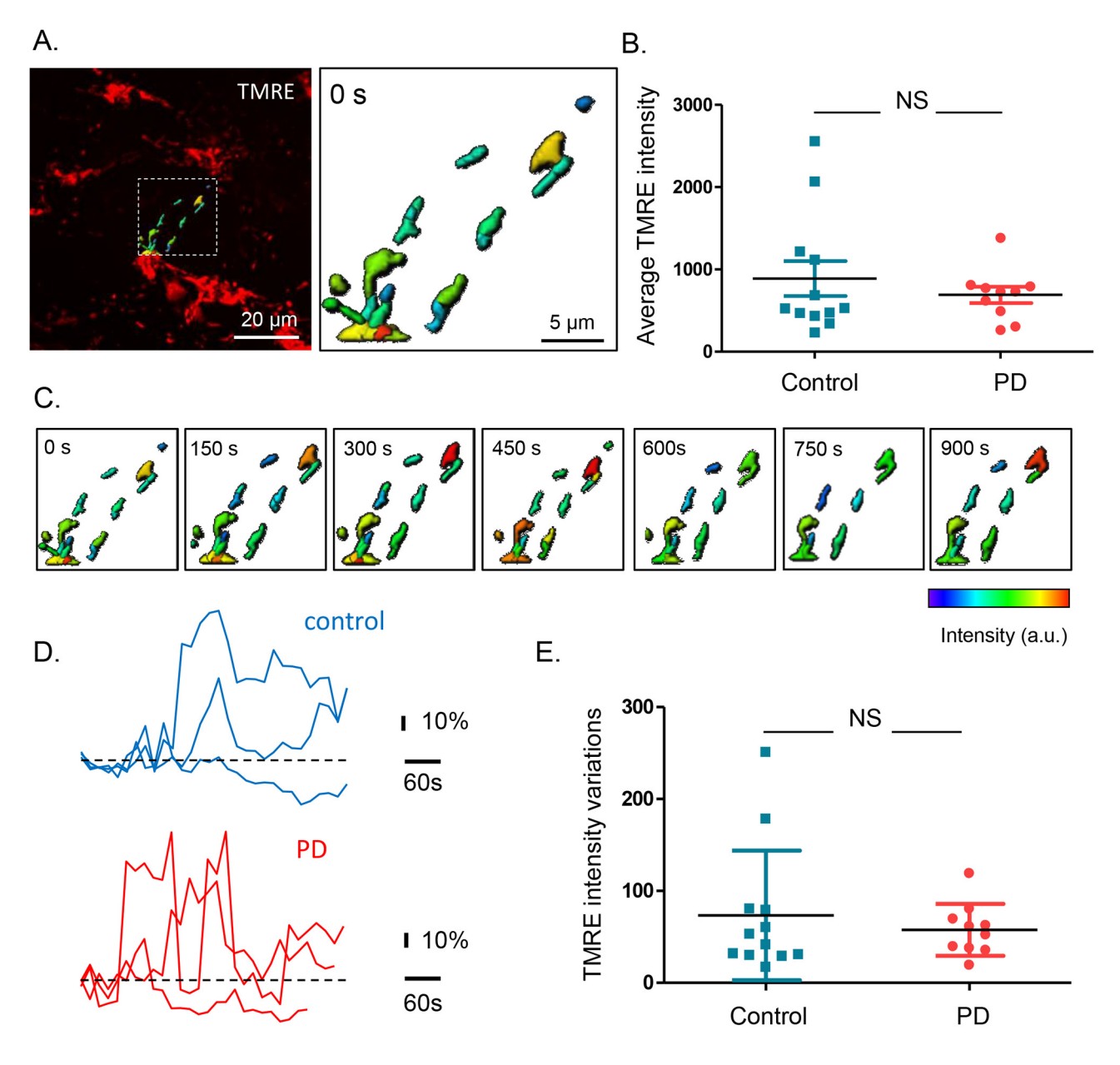

**Figure 3.** Mitochondrial membrane potential measurements in the submucous plexus of PD patients and controls. (**A.**) Representative example of a TMRE injection in the submucous plexus of a control subject. A subset of mitochondria was selected to show the TMRE intensity differences (as color-coded) in individual mitochondria. (**B.**) Summary data plot of the TMRE intensity in mitochondria of patients and controls (NS, not significantly different, p-value=0.49 (non-parametric Wilcoxon T-test). (**C.**) Example of a time series (900 s) of color-coded TMRE intensity fluctuations in individual mitochondria (see selection in **A.**). (**D.**) Graphs showing TMRE intensity variations (%) over time of three individual mitochondria of control and PD patient, suggesting similar dynamics in patients and controls. (**E.**) Summary data plot of the average TMRE intensity fluctuations for controls and PD patients. (NS, not significantly different, p-value=0.99, non-parametric Wilcoxon T-test).

The following source data and figure supplement are available for figure 3:

**Source data 1.** Mitochondrial membrane potential measurements.
**Figure supplement 1.** Mitochondrial TMRE destaining in the submucous plexus after addition of FCCP.

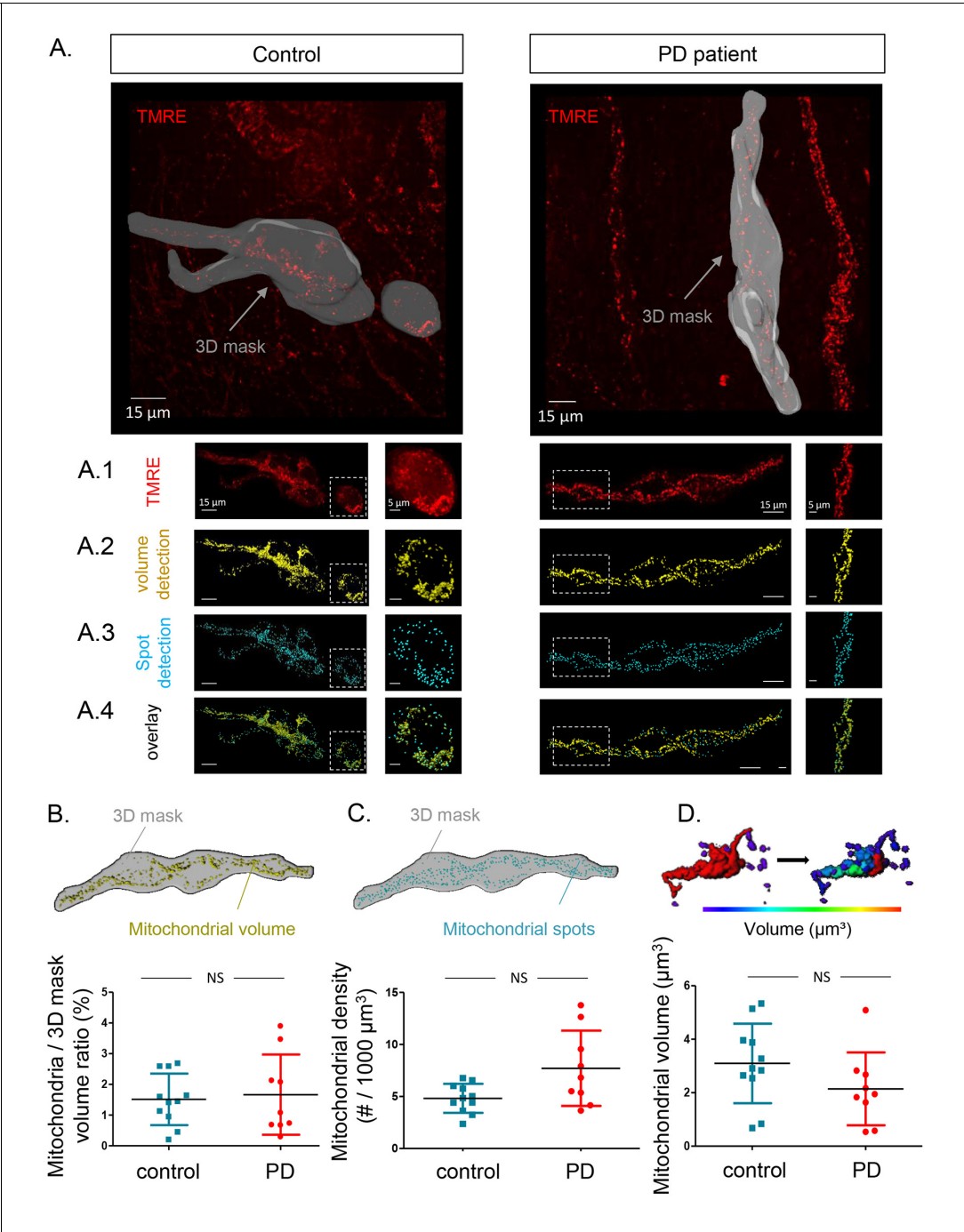

**Figure 4.** Mitochondrial number and volume in the submucous plexus of PD patients and controls. (**A.**) Representative example of a submucous plexus injected with TMRE for a control (left) and PD patient (right). The 3D mask drawn around the injected ganglion is shown in grey. The inside of this volume is enlarged in panels **A1-4** with an additional magnification of a selected subset (dashed square). Panels **A1-4** represent an example of mitochondrial volume detection (A2), mitochondrial spot detection (A3) and overlay of both (A4). (**B.**) Schematic representation of the total mitochondrial volume detection within the 3D mask and summary data plot of the volume (mitochondrial/3D mask) ratio quantification. (**C.**) Schematic representation of mitochondrial spot detection and summary data plot of the quantification (mitochondrial density (#/1000 $\mu m^3$). (**D.**) Schematic representation of the segmentation process (color-coded for mitochondrial size) to quantify the average volume ($\mu m^3$) of single mitochondria and a summary data plot of the quantification in controls versus patients. (NS, not significantly different, p-values (non-parametric Wilcoxon T-test); mitochondrial volume ratio = 0.65, mitochondrial density = 0.00.16, mitochondrial volume = 0.30).

The following source data is available for figure 4:

*Figure 4 continued*

**Source data 1.** Mitochondrial ratio.
**Source data 2.** Mitochondrial density.
**Source data 3.** Mitochondrial volume.

age, GI symptoms, disease duration or disease severity of PD patients (*Supplementary file 4*). We also tested whether α-synuclein aggregates were present in the samples using an antibody against α-synuclein. However, the staining patterns were indistinguishable between PD patients and controls (*Figure 6*).

## Discussion

In this study, we assessed the functionality of enteric neurons in PD patients and age-matched healthy controls using live imaging. Because PD patients often have GI problems, the ENS has been the subject of intensive study in the PD field for several decades (*Qualman et al., 1984*; *Kupsky et al., 1987*; *Wakabayashi et al., 1989*). A major driving force was the quest for a bio-marker to help diagnose PD in its premotor phase. In parallel, the hypothesis arose that the GI tract could be a starting point from where PD pathology propagates to the brain (*Braak et al., 2006*; *Hawkes et al., 2007*; *Pan-Montojo et al., 2010*, *2012*; *Li et al., 2008*, *2016*). Several groups have focused on immunohistochemical detection of α-synuclein aggregates in fixed GI tissue as a possible diagnostic biomarker for PD, with an emphasis on (sub)mucosal layers as these are accessible via endoscopy (*Shannon et al., 2012a*; *Braak et al., 2006*; *Wakabayashi et al., 1989*; *Hilton et al., 2014*; *Sánchez-Ferro et al., 2015*; *Lebouvier et al., 2010*; *Pouclet et al., 2012b*, *2012c*; *Shannon et al., 2012b*). So far, the outcome of these studies has been variable, possibly due to methodological differences (*Visanji et al., 2014*; *Ruffmann and Parkkinen, 2016*). Instead of searching for α-synuclein aggregation, only one study recently assessed mitochondrial morphology in the ENS of PD patients, but again using immunohistochemistry on fixed biopsies. We took a different approach and used live imaging to determine whether the submucous plexus in PD patients is functionally different from controls.

First, we found no significant differences in enteric neuronal $Ca^{2+}$ responses to various stimuli or in mitochondrial membrane potential, number and volume between the two populations. Second, numbers of neurons and ganglia in the biopsies were similar in PD patients and controls, indicating that the similar neuronal response patterns were not due to loss of the most vulnerable and dysfunctional neurons earlier in the disease. Last, we did not find differences in submucosal α-synuclein staining patterns between PD patients and controls.

Although GI symptoms were not an inclusion criterion for PD patients in this study, the PD patients had more GI symptoms than controls, in line with previous data (*Fasano et al., 2015*). Our finding of preserved neuronal functionality in these patients suggests that GI symptoms in PD do not arise from disturbed submucous neuronal function. Instead, GI symptoms may possibly be caused by impaired function of neurons of the myenteric plexus, the deeper nerve layer inner-vating the GI muscle. The submucous plexus pre-dominantly controls secretion, whereas the myenteric plexus predominantly controls motility

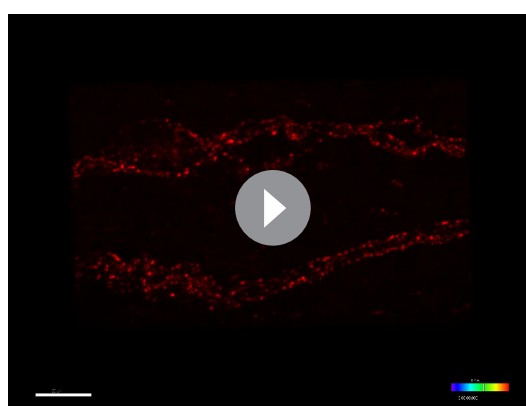

**Video 2.** Mitochondrial imaging in the human submucous plexus. TMRE-labeled mitochondria in a submucous ganglion of a control subject are shown as well as the selection of mitochondria that are color-coded for intensity (see also *Figure 3A*).

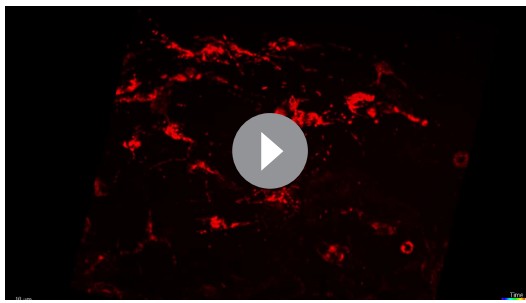

**Video 3.** Mitochondrial imaging in the human submucous plexus. Movie of TMRE-labeled mitochondria in a submucous ganglion of a PD patient. The TMRE intensity variations over time are shown as well as the 3D mask (in green, appearing half way in the movie) that is used to calculate the mitochondrial density in ganglia (see also *Figure 4A*).

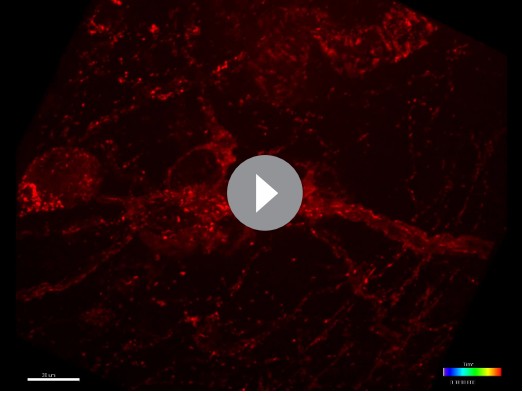

**Video 4.** Mitochondrial imaging in the human submucous plexus. Movie of TMRE-labeled mitochondria in a submucous ganglion of a control subject, showing TMRE fluctuations, selection of the ganglionic volume (3D mask) and mitochondrial volume detection (yellow) and spot detection (cyan) (see also *Figure 4* panels A1-4).

(*Furness, 2006*) and may thus be more involved in delayed gastric emptying and constipation in PD. However, it is not possible to safely sample the myenteric plexus with routine endoscopic biopsies because of the risk of bleeding. Another possible anatomic substrate for GI dysfunction in PD is the dorsal motor nucleus of the vagus in the brainstem, which is heavily affected by PD pathology (*Braak and Del Tredici, 2008*) and innervates neurons of the myenteric plexus via vagal nerve connections.

It has been suggested that the GI tract may be a site of initiation of PD. According to this hypothesis, an ingested pathogen may induce α-synuclein misfolding in submucous neurons, followed by retrograde axonal and transsynaptic propagation of α-synuclein misfolding via the vagus nerve to the brainstem (*Braak et al., 2006*; *Hawkes et al., 2007*). Chronic oral ingestion of low-dose rotenone in mice was reported to trigger α-synuclein accumulation in ENS ganglia and subsequently in the dorsal motor nucleus of the vagus nerve and substantia nigra, a sequence that was interrupted by vagotomy (*Pan-Montojo et al., 2010*, *2012*). Supporting this hypothesis, an epidemiological study reported that truncal vagotomy may be associated with a decreased risk of subsequent PD (*Tysnes et al., 2015*), although this link is still controversial (*Svensson et al., 2015*). Our data in living human neurons do not support this theory, as it seems unlikely that an ingested pathogen would induce toxicity in myenteric neurons and vagal nerve fibres while preserving the functionality of submucous neurons, which are located more closely to the gut lumen and whose fibres extend into the mucosa.

To our knowledge, this study is the first to investigate the functionality of living enteric neurons of patients with PD or any other neurodegenerative disease at the cellular and subcellular level. Many groups have modelled neurodegenerative diseases by generating neurons from patient fibroblasts via induced pluripotent stem cells (iPSCs) (*Ross and Akimov, 2014*), but it is still uncertain how faithfully iPSC-derived neurons mimic the behavior of endogenous primary neurons. Previous studies of the ENS in human PD have generally focused on patterns of α-synuclein immunoreactivity in fixed tissue. It should be kept in mind that the well-known Braak staging of PD is also exclusively based on detection of Lewy pathology but its link with neuronal function is still unclear (*Braak et al., 2003*). Ideally, future research into the progression of PD should also assess neuronal function and not just Lewy pathology.

Strengths of this study were its prospective design and the blinding of the investigators during imaging and data analysis. PD patients were clinically well-characterized, allowing us to search for correlations of cellular physiological and clinical parameters. Moreover, recruitment of the partners of PD patients as controls made it possible to make within-pair comparisons and minimize variability due to diet, lifestyle and other environmental factors. This is important in the light of recent data

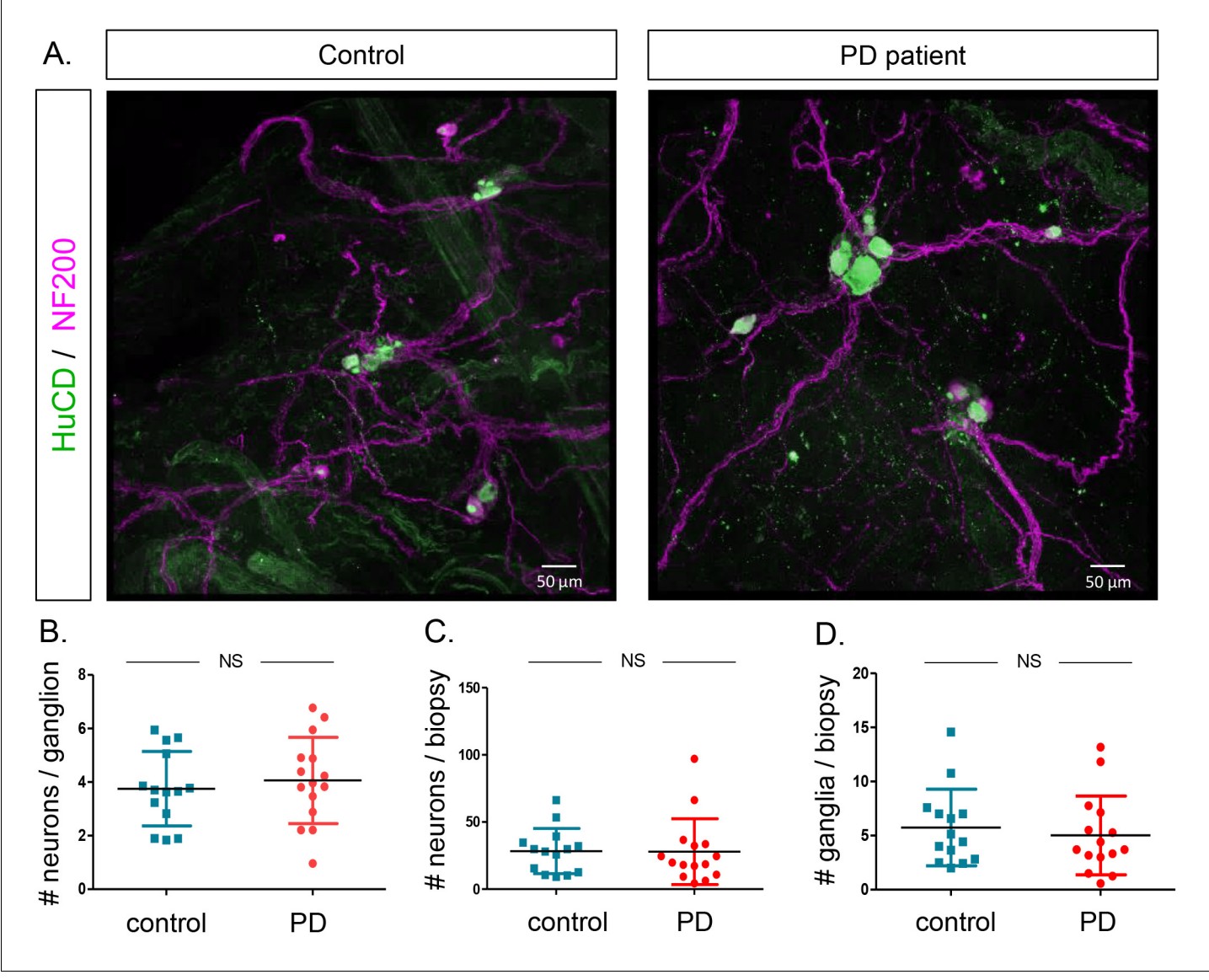

**Figure 5.** Quantification of neuron and ganglia numbers in the submucous plexus of PD patients and controls. (**A**.) Representative immunofluorescent staining of submucous plexus of a control and PD patient stained for the pan-neuronal marker HuCD (green) and neuronal filament marker NF200 (Magenta). Bars: 50 µm. (**B**.) Graph showing quantification of number of neurons per ganglion. (**C**.) Graph showing quantification of total number of neurons per biopsy. (**D**.) Graph showing quantification of number of ganglia per biopsy. (NS, not significantly different (non-parametric Wilcoxon T-test), p-values; # neurons/ganglion = 0.86, # neurons/biopsy = 0.17 (non-parametric Wilcoxson T-test), # ganglia/biopsy = 0.24).

The following source data is available for figure 5:

**Source data 1.** Number of neurons per ganglia.
**Source data 2.** Number of neurons per biopsy.
**Source data 3.** Number of ganglia per biopsy.

showing that cohabitation results in overlapping gut microbiome profiles (*Yatsunenko et al., 2012*) and that the composition of the gut microbiome may be related to the clinical manifestations of PD (*Scheperjans et al., 2015*).

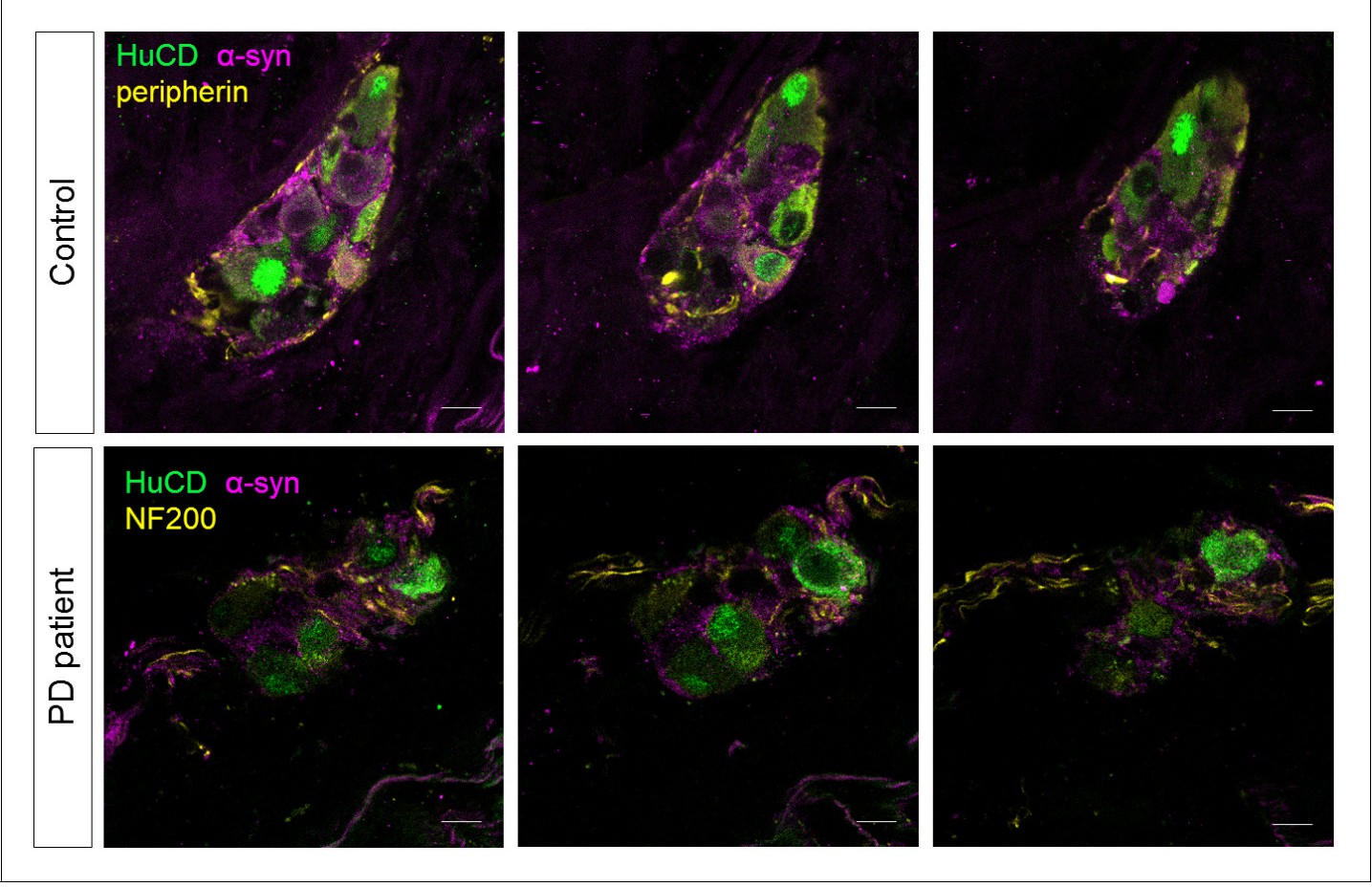

**Figure 6.** α-synuclein staining in the submucous plexus of PD patients and controls. Top and bottom rows show three single slice confocal and deconvolved images of a representative immunofluorescent staining in the submucous plexus of a control (top) and PD patient (bottom) stained for the pan-neuronal marker HuCD (green) and α-synuclein (magenta). A neuronal fiber staining (yellow: NF200 in bottom and peripherin in top row) was added to help delineating the submucous ganglion. No differences in α-synuclein staining patterns in PD patients and controls could be detected. Bars: 10 μm.

Our study also has some limitations. The number of subjects was relatively small. Power calculations for this study were difficult because live imaging of PD enteric neurons has never been performed before and no data were available to reliably estimate expected differences and standard deviations. Patients with young-onset PD were somewhat overrepresented, possibly reflecting the greater willingness of young PD patients to participate in a demanding study. Another limitation was that the PD and control groups were not gender-matched. This was a consequence of the pairwise recruitment in combination with the fact that the majority of PD participants, as in most clinical PD studies, were male. Another criticism could be that the $Ca^{2+}$ and mitochondrial imaging assays may not be sensitive enough to detect subtle impairments in neuronal function. Nevertheless, using the same technology, we previously found alterations in submucous neurons in functional dyspepsia patients (*Cirillo et al., 2015*), demonstrating the robustness of the techniques to detect changes, even subtle, in disease conditions. Another limitation is that we were unable to sample and analyze myenteric neurons due to safety reasons, as discussed above. It is also possible that the duodenum was not the optimal site for detection of neuronal changes in PD. However, we precisely chose a proximal GI site based on evidence that α-synuclein aggregation in the ENS in PD is deposited with a rostrocaudal gradient (*Visanji et al., 2014*). Moreover, proximal portions of the GI tract receive stronger vagal innervation than distal regions (*Ruffmann and Parkkinen, 2016*) which, in view of the hypothesis that the vagus nerve transmits PD pathology, makes this region especially interesting.

Also, duodenal endoscopy is better tolerated and requires less demanding patient preparation compared to colonoscopy. Finally, we cannot exclude the possibility that the duodenal submucosal plexus in PD patients is affected in a non-uniform, 'patchy' fashion and that we missed affected ganglia due to spatial sampling error.

In conclusion, we have applied live imaging techniques to investigate neuronal physiology in primary neurons from PD patients. Our findings suggest that GI symptoms in PD do not arise from dysfunction of submucous neurons. Furthermore, our data do not support the theory that the disease process of PD initiates in the submucous layers of the ENS and spreads from there to the brain.

## Materials and methods

### Study population

Fifteen patients with PD and fifteen healthy age-matched controls were recruited in pairs in the Movement Disorders Clinic of University Hospitals Leuven. Each pair consisted of a PD patient and his/her healthy partner. We recruited pairwise in order to allow within-pair comparisons and minimize variability due to diet, lifestyle and other environmental factors. Inclusion criterion for PD patients was diagnosis of PD according to the Gelb criteria (*Gelb et al., 1999*). GI symptoms were not a requirement for inclusion. Exclusion criteria for PD patients were: cognitive impairment that, in the opinion of the treating neurologist, interfered with the ability to fully understand the patient information brochure; and presence of GI disorders unrelated to PD. Exclusion criteria for the controls were GI and neurological diseases. All PD and control subjects completed the SCOPA-AUT, a questionnaire designed to assess autonomic (including GI) symptoms in PD patients (*Visser et al., 2004*), within one month before the endoscopic procedure. On the morning of the gastroduodenoscopy disease severity was assessed in the PD patients by a movement disorders neurologist by means of UPDRS part III and Hoehn-Yahr (HY) scale in practically defined 'off' state, that is, at least 12 hr after the last intake of medication. UPDRS parts I, II and IV and Mini Mental State Examination (MMSE) were also completed. Levodopa-equivalent daily dose (LED) was calculated as described (*Tomlinson et al., 2010*). Disease duration was based on the onset of the first motor symptom as reported by the patient. On the morning of the procedure, control subjects were also clinically assessed by the movement disorders neurologist to exclude parkinsonism. The ethics committee of the University Hospitals Leuven approved the study and all subjects gave written informed consent and consent to publish according to the declaration of Helsinki.

### Gastroduodenoscopy and biopsy preparation

An experienced endoscopist at the Gastroenterology unit of the University Hospitals Leuven obtained 8 biopsies from the second part of the duodenum from each PD patient and control. Each pair underwent endoscopy on the same day immediately one after another in random order. All subjects had a macroscopically normal upper GI tract, except for one control with minor reflux esophagitis. The duodenal biopsies were immediately immersed in oxygenated ice-cold Krebs solution (in mM: 120.9 NaCl, 5.9 KCl, 1.2 $MgCl_2$, 2.5 $CaCl_2$, 11.5 glucose, 14.4 $NaHCO_3$, and 1.2 $NaH_2PO_4$) and coded. All subsequent tissue manipulations, experiments and data analyses were performed by investigators blinded to the disease status (PD versus control) of the subject. The submucous plexus was removed from the mucosal epithelium by microdissection, as described previously (*Cirillo et al., 2013*), and used the same day for live calcium ($Ca^{2+}$) or mitochondrial imaging. A schematic overview of the experimental workflow is presented in *Figure 1*.

### Calcium imaging

Submucous plexus preparations (2 per subject) were loaded at RT for 20 min with the fluorescent $Ca^{2+}$ indicator Fluo-4 AM (1 µM, Molecular Probes, Merelbeke, Belgium) in Krebs buffer containing 0.01% Cremophor EL surfactant agent (Fluka Chemika, Buchs, Switzerland). After rinsing, tissues were imaged as previously described (*Cirillo et al., 2013*).

To elicit neuronal activity, we used three different stimuli: first, a brief high $K^+$ (10 s, 75 mM) induced depolarization was applied via a local perfusion pipette to induce a sharply rising $Ca^{2+}$ transient and test neuron viability. Second, fibre tracts were electrically stimulated by trains (2 s, 20 Hz) of 300 µs electrical pulses (Grass Instruments, Rhode Island, USA) applied via a tungsten electrode

(diameter 50 µm). Third, the nicotinic cholinergic receptor agonist DMPP (10 µM, Fluka Chemika, Buchs, Switzerland) was locally perfused for 20 s.

Images were collected using Till Vision software (TILL Photonics, Gräfelfi, Germany) and analysis was performed using custom-written macros in IGOR PRO (Wavemetrics, Lake Oswego,OR). To remove drift and movement artefacts due to perfusion, the image stack was registered to the first image. Regions of interest (ROIs) were drawn over each neuron and fluorescence intensities were calculated, normalized and expressed as an $F/F_0$ ratio (with $F_0$ being the baseline fluorescence). Transient $[Ca^{2+}]_i$ peaks were considered if they exceeded the baseline plus 5 times the intrinsic noise level. The percentage of responsive cells was determined. The maximum $[Ca^{2+}]_i$ peak amplitude was calculated as a percentage change above baseline.

## Mitochondrial imaging

Submucous plexus preparations (3 per subject) were pinned flat in a Sylgard (Dow Corning) containing dish. A glass microinjection capillary (pulled on a P87 Sutter Instruments pipette puller) filled with TMRE (300 nM, Thermo Fischer, Merelbeke, Belgium) was navigated into a submucous ganglion using a pneumatic manipulator (Narishige, New York, USA). Local injection of the mitochondrial dye was essential in order to avoid background loading of connective tissues and cellular structures other than those of the submucous plexus. This approach differs from TMRE loading protocols in cellular monolayers (*O'Reilly et al., 2003*; *Perry et al., 2011*). However, even though the TMRE concentration used in the injection pipette is relatively high, we assume that all observations were made in non-quenching TMRE mode, because after the topical injection (~150 nl) the dye rapidly diffuses into a larger volume (dilution by a factor 40 if the volume of biopsy [6 mm$^3$], or ~3.10$^3$ if the recording bath volume 500 µl is considered). Moreover, image stacks were recorded in SMP structures away from the injection spot, to assure we recorded at lower concentrations than what was injected.

To test whether TMRE specifically labelled mitochondria, we used carbonyl cyanide-4-(trifluoro-methoxy)phenylhydrazone (FCCP, 3 µM), a protonophore that quickly dissipates mitochondrial membrane potential. The complete destaining indicates that TMRE specifically labeled mitochondria (*Figure 3—figure supplement 1*).

In an earlier preliminary set of experiments, Mitotracker green (10 µM, Life Technologies, Merelbeke, Belgium) was also injected in an attempt to determine mitochondrial volumes in a membrane potential-independent way. However, Mitotracker green did not diffuse from the injection spot and there was no spread of the dye in human tissue. The reason is unclear but most likely higher temperatures (37°C) are needed for the dye to spread uniformly. We therefore abandoned the Mitotracker green strategy and focused on single dye (TMRE) injections, and used the TMRE signals at RT to determine mitochondrial volume and density.

After injection, the tissues were imbedded in 1.5% low-melting point agarose dissolved in Krebs buffer, which allowed transfer of the preparations close and flat enough to a glass coverslip to be recorded from on an inverted spinning disk confocal microscope (Nikon Ti - Andor Revolution - Yokogawa CSU-X1 Spinning Disk (Andor, Belfast, UK)) with a Nikon 40x lens (LWD, NA 1.1, WI). For fast 3D stacks we used a Piezo Z Stage controller, and recorded both at RT (to analyze the morphology of the mitochondria) and at physiological temperatures (37°C) (to analyze dynamic mitochondrial membrane potential fluctuations). Image stacks were deconvolved using a theoretical point spread function based on the optical properties of the imaging system (pinhole spacing (6.33 µm) and back-projection radius (625 nm)) and stabilized using Huygens professional (SVI, Hilversum, The Netherlands). The background fluorescence was automatically estimated and corrected for using Huygens' default parameters as well as (when necessary) photo-bleaching, always assuming the first image to be the brightest.

Subsequently, the deconvolved image stacks were imported in IMARIS 8.0.1 (Bitplane, Zurich, Switzerland) to assess mitochondrial volume and intensity characteristics. First, a volume of interest was drawn around the injected ganglion, which was then used as a 3D mask, within which the spot and volume detection (absolute intensity thresholding) algorithms available in IMARIS were applied. Also mitochondrial intensity changes over time were analysed in the image stacks recorded at 37°C by tracking mitochondria over time. In this analysis, 500 mitochondria per volume were selected from the output of Imaris' spot detection algorithm, which sorted detected spots based on maximum intensity in the spot centre.

## Immunohistochemistry

After Ca$^{2+}$ imaging and mitochondrial analysis, the submucous plexus preparations were fixed for 30 min in 4% paraformaldehyde (PFA, Merck, Overijse, Belgium) for post-hoc immunohistochemical confirmation of neuronal identity. In addition, the submucous plexus was isolated from three fresh biopsies per subject (without live imaging) and immediately fixed in PFA for immunohistochemical analysis of numbers of ganglia, neurons per ganglion and numbers of non-ganglionic (individual) neurons, as described earlier (*Cirillo et al., 2013*).

Primary antibodies against two typical enteric neuron markers: neurofilament NF200 (chicken anti-NF200 1/50000; Abcam, Cambridge, UK, RRID: AB_2149618) and HuCD (mouse anti-HuCD 1/500; Molecular Probes, Merelbeke, Belgium, RRID: AB_221448), were used after 2 hr (at RT) in blocking buffer containing 0.5% Triton X-100 (Thermo Fischer, Merelbeke, Belgium) and 4% serum matched to the host of the secondary antibody. After three cycles of washing (PBS), fluorescently labelled secondary antibodies were then added for 2 hr (at RT). After final washing (PBS), tissues were mounted on a microscope slide in citifluor (Citifluor Ltd.,Leicester,UK). Confocal images were recorded using a Zeiss LSM 780 confocal microscope (Zeiss, Belgium). Additionally, α-synuclein (SC-7011-R, Santa Cruz Biotechnology, Dallas, Texas, US, RRID:AB_2192953) antibodies were used on one fixed submucous plexus of each subject. Confocal images and blinded analysis were performed to evaluate labelling patterns and possible aggregation in these tissues.

## Statistical analysis

All experiments and analyses were performed in a blinded fashion. Investigators were unblinded only after the analysis of imaging data for all subjects had been finalized. All results are presented as mean ± SD, except for Hoehn-Yahr scores, which are presented as median and interquartile ranges. Differences between groups were analysed using paired tests (for details see below). Imaging parameters were also correlated with clinical characteristics (age, SCOPA GI symptoms, UPDRS III off, disease duration) using linear correlations. Based on data distributions, there was no reason to assume any higher order relations. Non-parametric (Wilcoxon test/Spearman correlation) tests were used based on the outcome of Shapiro-Wilk tests for normal data distribution. A Bonferroni correction was used to correct for multiple testing.

## Acknowledgements

We like to thank Maura Corsetti and Tim Vanuytsel for the help during gastroduodenoscopy. All imaging was performed on the microscopes of LENS and CIC and we like to thank NVIDIA for donating a K40 GPU card. We thank the members of LENS and Natalia Pessoa Rocha for their assistance during the recording days and for their constructive comments on the project and manuscript.

## Additional information

### Funding

| Funder | Grant reference number | Author |
| --- | --- | --- |
| University of Leuven | GOA/13/017 | Wim Vandenberghe Pieter Vanden Berghe |
| Fonds Wetenschappelijk Onderzoek | Hercules AKUL/09/050 | Pieter Vanden Berghe |
| Fonds Wetenschappelijk Onderzoek | G.0A44 | Wim Vandenberghe Pieter Vanden Berghe |
| Fonds Wetenschappelijk Onderzoek | 3M130239 | An-Sofie Desmet |
| Fonds Wetenschappelijk Onderzoek | 3M120390 | Carla Cirillo |

The funders had no role in study design, data collection and interpretation, or the decision to submit the work for publication.

## Author contributions

A-SD, Formal analysis, Funding acquisition, Validation, Investigation, Visualization, Methodology, Writing—original draft, Project administration, Writing—review and editing; CC, Conceptualization, Formal analysis, Funding acquisition, Validation, Investigation, Visualization, Methodology, Writing—original draft, Project administration, Writing—review and editing; JT, Resources, Supervision, Funding acquisition, Methodology, Writing—review and editing; WV, Conceptualization, Resources, Supervision, Funding acquisition, Project administration, Writing—review and editing; PVB, Conceptualization, Software, Formal analysis, Supervision, Funding acquisition, Validation, Investigation, Visualization, Methodology, Writing—original draft, Project administration, Writing—review and editing

## Author ORCIDs

An-Sofie Desmet, http://orcid.org/0000-0003-1885-0815
Carla Cirillo, http://orcid.org/0000-0002-8445-5955
Wim Vandenberghe, http://orcid.org/0000-0002-9758-5062
Pieter Vanden Berghe, http://orcid.org/0000-0002-0009-2094

## Ethics

Human subjects: The ethics committee of the University Hospitals Leuven approved the study and all subjects gave written informed consent according to the declaration of Helsinki.

## Additional files

### Supplementary files

• Supplementary file 1. Clinical characteristics of individual PD patients. Clinical data of individual PD patients. SCOPA total, *Patient six entered 'not applicable' for the 2 SCOPA items related to sexual function. Mean ± SD are shown. For Hoehn-Yahr scores, median and interquartile range are shown instead of average and SD.

• Supplementary file 2. Correlation between $Ca^{2+}$ imaging data and PD characteristics Spearman R-values of correlations between $Ca^{2+}$ imaging parameters and clinical characteristics of PD patients (gray shaded rows) and where applicable (age, SCOPA) of controls (white rows).

• Supplementary file 3. Correlations between mitochondrial imaging data and PD characteristics Spearman R-values of correlations between mitochondrial imaging parameters and clinical characteristics of PD patients (gray shaded rows) and where applicable (age, SCOPA) of controls (white rows).

• Supplementary file 4. Correlations between immunohistochemical data and PD characteristics Spearman R-values of immunofluorescent counting correlated with clinical characteristics of the PD patients (gray shaded rows) and where applicable (age, SCOPA) of controls (white rows).

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
