## [Decision Letter]

Thank you for submitting your article "Live calcium and mitochondrial imaging in the enteric nervous system of Parkinson patients and controls" for consideration by *eLife*. Your article has been favorably evaluated by a Senior Editor and four reviewers, one of whom is a member of our Board of Reviewing Editors. The reviewers have opted to remain anonymous.

The reviewers have discussed the reviews with one another and the Reviewing Editor has drafted this decision to help you prepare a revised submission.

Please address the experimental concerns of reviewer #3 and the other concerns of all the reviewers by editing the text. For example, as noted by several reviewers, the authors should clearly state that a limitation of their study is that they did not examine the myenteric plexus, unless they can provide literature support that the submucosal plexus they sampled has been shown previously to display α-synuclein aggregates. Reviewers #2 and #4 point about "patchiness" is also important to consider and comment on.

The revisions will be assessed by the handling Reviewing Editor but won't necessarily be evaluated by all of the original reviewers.

*Reviewer #1:*

Using powerful live cell imaging methods of control and PD patient duodenum biopsies, Desmet and colleagues find that calcium handling, mitochondrial functions and number and a-synuclein immunostaining are normal in submucosal enteric neurons. The methods and data collected seem clear and the manuscript is very well written. The main problem is that myenteric plexus neurons, that are implicated in the constipation phenotype of PD patients, lie too deep for analysis. Thus, the submucosal plexus results are less that definitive evidence arguing against several gut to brain hypotheses. Nevertheless, the results do impact important issues regarding PD diagnosis and etiology.

*Reviewer #2:*

Gastrointestinal dysfunction is a prominent non-motor feature of Parkinson's disease (PD). The alimentary tract as a whole is affected, with abnormal salivation, dysphagia, delayed gastric emptying, constipation and defecatory dysfunction. The underlying mechanisms are poorly understood, but it has been suggested that they may have something to do with the abnormal aggregation of α-synuclein in the enteric nervous system. The presence of Lewy pathology in the enteric nervous system in PD has been known since 1984. It is found in both Auerbach's and Meissner's plexuses. One school of thought believes that α-synuclein-containing inclusions first appear in the enteric nervous system, from where they progress to brain regions, such as the dorsal motor nucleus of the vagus nerve and the substantia nigra. A pathogen may penetrate the gut epithelium and enter axons of the enteric neurons in the myenteric plexus (Auerbach's plexus), which controls the activity of the smooth muscles of the gut and/or axons of the submucosal plexus (Meissner's plexus), which regulates mucosal secretion and blood flow. The present study used live imaging of neurons from the duodenum, of patients with PD and controls obtained through gastroduodenoscopy. All the patients with PD were undergoing treatment with oral dopaminergic medication. The authors investigated calcium signalling properties of submucous neurons, mitochondrial membrane potentials of these neurons, mitochondrial volumes and numbers, as well as numbers of nerve cells. No differences with controls were found, nor were there any α-synuclein aggregates. This study, which used a reasonable number of patients with PD and several innovative techniques, suggests that submucosal neurons of the duodenum were not functionally impaired and that gastrointestinal dysfunction in these patients must have had other reasons. It is widely believed that it is the myenteric plexus which mostly controls motility and dysfunction of which thus could thus play a role in constipation in PD. The absence of α-synuclein aggregates in endoscopic biopsies that do not include the myenteric plexus should be interpreted with caution. One must also bear in mind that these patients had clinical PD, with abundant α-synuclein inclusions in brain. What one cannot exclude is that the enteric nervous system was affected during the prodrome of PD, but did not have any more α-synuclein inclusions in the clinical phase. Of course, this study is limited by the fact that the authors only looked at duodenal biopsies. There might have been α-synuclein inclusions elsewhere. Is it possible that there were some in the duodenum as well, but that they were missed during gastroduodenoscopy?

*Reviewer #3:*

The manuscript by Desmet et al. describes an important set of experiments examining the properties of live, human submucosal neurons taken from Parkinson's disease (PD) patients and spousal controls. These studies found no differences between PD submucosal neurons and controls in several assays, including apparent viability, neuronal density, cytosolic Ca^2+^ influx in response to depolarization, mitochondrial membrane potential, mitochondrial density or α-synuclein immunoreactivity. Given the widespread belief that PD originates in the GI tract and initially manifest as submucosal pathology, the results of this study, even though they are negative, are very important. From a technical standpoint, the studies are very nicely done and clearly illustrated. The manuscript is well-written (although the Discussion should be shortened), circumspect (e.g., the limitations of the study are clearly stated) and scholarly. I have only a few concerns.

• The TMRE measurements should be calibrated by using oligomycin and FCCP to hyperpolarize and depolarize mitochondria. Was TMRE at non-quenching concentrations?

• Using TMRE for the mitochondrial density estimates is a bit problematic because it brings mitochondrial membrane potential into the picture, whereas a mitochondrially targeted GFP or mitotracker green wouldn't have done so. I understand the need for a rapid, diffusible marker with the biopsies but the authors should discuss the limitations of this aspect of the study in the main body of the text.

• Was there any difference in mitochondrial morphology in the PD and control samples?

*Reviewer #4:*

Desmet et al. investigate enteric neuronal function from duodenal biopsies in patients with Parkinson's disease and controls. The goal is to explore whether functional assays can detect evidence consistent with the established increased prevalence of GI symptomatology and predilection for α-synuclein pathology in the enteric nervous system. The experiments do not detect any differences between cases and spouse controls, based on calcium imaging, mitochondrial membrane potential, and assessments of mitochondrial numbers. None of the biopsies had demonstrable α-synuclein aggregate pathology, making it difficult to conclude whether the negative results might be due to sampling issues. While innovative, the negative results in this small cohort, allow only preliminary conclusions to be drawn at this stage. This manuscript is not appropriate for *eLife*, and would be better suited to a neurology journal.

Suggestions for improving the manuscript:

It would be helpful to discuss further whether prior studies have found synuclein pathology with similar frequency in the submucosal vs. myenteric plexus.

Consider the possibility that pathology and dysfunction of the ENS may be "patchy", complicating interpretation of studies of selective biopsies.

It would be interesting in the Discussion (or Introduction) to discuss any results from PD animal model studies of the GI tract.

The discussion could be shortened, and suggest caution not to overstate the conclusion given the many caveats: "our findings… strong suggests that GI symptoms in PD do not arise from disturbed sub mucous neuronal function".

The authors also overstate the degree of controversy surrounding enteric nervous system pathology in PD (e.g. Introduction, second paragraph). While there is certainly debate about the potential utility as a clinical biomarker and the best staining protocol, I believe most experts agree that the ENS is pathologically involved.

---

## [Author Response]

*Reviewer #1:*

*Using powerful live cell imaging methods of control and PD patient duodenum biopsies, Desmet and colleagues find that calcium handling, mitochondrial functions and number and a-synuclein immunostaining are normal in submucosal enteric neurons. The methods and data collected seem clear and the manuscript is very well written. The main problem is that myenteric plexus neurons, that are implicated in the constipation phenotype of PD patients, lie too deep for analysis. Thus, the submucosal plexus results are less that definitive evidence arguing against several gut to brain hypotheses. Nevertheless, the results do impact important issues regarding PD diagnosis and etiology.*

As the myenteric plexus lies too deep to safely sample during endoscopy, our study indeed does not provide data on myenteric neurons. We agree with the reviewer that this is an important limitation. We already indicated this in the Discussion of the previous version of the manuscript and included this again in the revised manuscript: ‘GI symptoms may possibly be caused by impaired function of neurons of the myenteric plexus, the deeper nerve layer innervating the GI muscle. The submucous plexus predominantly controls secretion, whereas the myenteric plexus predominantly controls motility, and may thus be more involved in delayed gastric emptying and constipation in PD. However, it is not possible to safely sample the myenteric plexus with routine endoscopic biopsies because of the risk of bleeding.’

To emphasize this limitation even more strongly, we have added the following statement to the Discussion: ‘Another limitation of the study is that we were unable to sample and analyze myenteric neurons due to safety reasons, as indicated above.’

Nevertheless, we believe that our functional data on submucosal neurons are very relevant in the context of PD, even in the absence of myenteric plexus data, for three reasons:

1) The submucosal plexus is located more closely to the gut lumen than the myenteric plexus. Considering the hypothesis that PD may be triggered by environmental pathogens that enter via the lumen, submucosal neurons would be more likely to be affected than myenteric neurons.

2) Submucosal and myenteric neurons are synaptically and functionally connected. Based on the hypothesis of transsynaptic spread of pathological forms of α-synuclein, it would seem unlikely that one of these layers would be severely affected while the other would be intact.

3) Several publications have reported abnormal α-synuclein aggregates in the submucosal plexus of PD patients (references 1-6 below). In fact, we have not found any evidence in the immunohistochemical literature suggesting that pathology in PD might selectively affect the myenteric plexus while sparing the submucosal plexus.

*Reviewer #2:*

*Gastrointestinal dysfunction is a prominent non-motor feature of Parkinson's disease (PD). The alimentary tract as a whole is affected, with abnormal salivation, dysphagia, delayed gastric emptying, constipation and defecatory dysfunction. The underlying mechanisms are poorly understood, but it has been suggested that they may have something to do with the abnormal aggregation of α-synuclein in the enteric nervous system. The presence of Lewy pathology in the enteric nervous system in PD has been known since 1984. It is found in both Auerbach's and Meissner's plexuses. One school of thought believes that α-synuclein-containing inclusions first appear in the enteric nervous system, from where they progress to brain regions, such as the dorsal motor nucleus of the vagus nerve and the substantia nigra. A pathogen may penetrate the gut epithelium and enter axons of the enteric neurons in the myenteric plexus (Auerbach's plexus), which controls the activity of the smooth muscles of the gut and/or axons of the submucosal plexus (Meissner's plexus), which regulates mucosal secretion and blood flow. The present study used live imaging of neurons from the duodenum, of patients with PD and controls obtained through gastroduodenoscopy. All the patients with PD were undergoing treatment with oral dopaminergic medication. The authors investigated calcium signalling properties of submucous neurons, mitochondrial membrane potentials of these neurons, mitochondrial volumes and numbers, as well as numbers of nerve cells. No differences with controls were found, nor were there any α-synuclein aggregates. This study, which used a reasonable number of patients with PD and several innovative techniques, suggests that submucosal neurons of the duodenum were not functionally impaired and that gastrointestinal dysfunction in these patients must have had other reasons. It is widely believed that it is the myenteric plexus which mostly controls motility and dysfunction of which thus could thus play a role in constipation in PD. The absence of α-synuclein aggregates in endoscopic biopsies that do not include the myenteric plexus should be interpreted with caution.*

See our response to reviewer 1.

*One must also bear in mind that these patients had clinical PD, with abundant α-synuclein inclusions in brain. What one cannot exclude is that the enteric nervous system was affected during the prodrome of PD, but did not have any more α-synuclein inclusions in the clinical phase.*

It is important to note that we did not find any difference in numbers of submucosal neurons between PD patients and controls. This argues against the possibility that the observed lack of functional or immunocytochemical abnormalities in the submucosal plexus of PD patients was due to loss of the most vulnerable and affected submucosal neurons in earlier phases of the disease (Discussion, second paragraph). We cannot entirely rule out the possibility that some submucosal neurons may have been dysfunctional and/or may have contained α-synuclein inclusions in the prodromal phase of PD but functionally recovered and cleared their inclusions later on in the course of the disease. However, this possibility seems extremely unlikely, given the progressive neurodegenerative nature of PD.

*Of course, this study is limited by the fact that the authors only looked at duodenal biopsies. There might have been α-synuclein inclusions elsewhere.*

We agree with this point. We already discussed this limitation in the previous version of the manuscript and included this again in the revised manuscript): ‘It is possible that the duodenum was not the optimal site for detection of neuronal changes in PD. […] Also, duodenal endoscopy is generally better tolerated and requires less demanding patient preparation compared to colonoscopy.’

*Is it possible that there were some in the duodenum as well, but that they were missed during gastroduodenoscopy?*

Indeed, we cannot exclude the possibility that the duodenal submucosal plexus in PD patients may be affected in a ‘patchy’ fashion and that we missed affected ganglia due to sampling error. To acknowledge this, we have added the following sentence to the Discussion: “Finally, we cannot exclude the possibility that the duodenal submucosal plexus in PD patients is affected in a non-uniform, ‘patchy’ fashion and that we missed affected ganglia due to spatial sampling error.”

*Reviewer #3:*

*The manuscript by Desmet et al. describes an important set of experiments examining the properties of live, human submucosal neurons taken from Parkinson's disease (PD) patients and spousal controls. These studies found no differences between PD submucosal neurons and controls in several assays, including apparent viability, neuronal density, cytosolic Ca^2+^ influx in response to depolarization, mitochondrial membrane potential, mitochondrial density or α-synuclein immunoreactivity. Given the widespread belief that PD originates in the GI tract and initially manifest as submucosal pathology, the results of this study, even though they are negative, are very important. From a technical standpoint, the studies are very nicely done and clearly illustrated. The manuscript is well-written (although the Discussion should be shortened), circumspect (e.g., the limitations of the study are clearly stated) and scholarly. I have only a few concerns.*

*• The TMRE measurements should be calibrated by using oligomycin and FCCP to hyperpolarize and depolarize mitochondria.*

We agree that the fluorescence generated by TMRE needs to be interpreted with caution and that is indeed important to pharmacologically check whether the labeling is targeted specifically to mitochondria. To this end we used, as suggested, FCCP to fully depolarize mitochondria. We found that there was a complete destaining, which indeed indicates that our loading paradigm specifically labeled mitochondria. We included this in the Materials and methods section and added a figure of a destaining experiment as a supplementary figure in the revised manuscript Figure 3—figure supplement 1.

Due to the complex nature of the biopsy preparations, we were not able to fully calibrate the TMRE signals using consecutive additions of oligomycin and FCCP. The main reason is that our loading protocol differed substantially from what is used in monolayer cell cultures. Instead of loading TMRE uniformly during tens of minutes, we chose to topically inject TMRE in the sparse neuronal structures present in the biopsy. In this way, we avoided labeling of mitochondria in other than submucous ganglionic structures. For the same reason, we did not add a low concentration of TMRE in the recording buffer as needed for calibration (references 7, 8 below). This decision precluded TMRE calibration in our preparations as one of the necessary steps for calibration is to hyperpolarize the mitochondria using oligomycin. In the absence of TMRE in the surrounding buffer, hyperpolarization would not cause uptake of extra dye. Even though in this study we are not able to derive absolute mitochondrial membrane potentials (in mV) from the TMRE fluorescence, we are convinced that the signals correctly reflect the mitochondrial potentials in the two groups and that our comparison between PD patients and control subjects is entirely valid.

*Was TMRE at non-quenching concentrations?*

In preliminary experiments, we optimized the injection methodology as well as the TMRE concentration. We found that ± 10 minutes after injection of a local bolus of TMRE (at a relatively high concentration of 300 nM) images with a good signal to noise ratio could be recorded. This loading procedure varies substantially from what is possible in cellular monolayers (7,8), where often the dye (albeit at a lower concentrations) is incubated for several minutes (see also comment related to TMRE calibration). However, even though the TMRE concentration used in the injection pipette is relatively high, we assume that all observations were made in non-quenching TMRE mode, because after the topical injection (~ 150 nl^**^) the dye rapidly diffuses into a larger volume (dilution by a factor 40 if the volume of biopsy [6mm³], or ~3.10^3^ if the 500 µl recording bath volume is considered).

Moreover, image stacks were recorded in submucosal plexus structures away from the injection spot, to assure we recorded at lower concentrations than what was injected. The assumption that the TMRE recordings were made in non-quenching mode is corroborated by the fact that closer to the injection site mitochondria behaved similarly than at a short distance away. Therefore, we are convinced that the signals can be reliably interpreted as a good qualitative assessment of mitochondrial function.

We have expanded the Materials and methods section to describe the loading methodology, including the discussion about dye concentration and diffusion after injection (subsection “Mitochondrial imaging”, first paragraph).

^**^ We measured the ejected volume by navigating the pipette tip (n=4) in a drop of oil after which the volume of the drop could be estimated based on the diameter of the sphere (160 ± 47 nl).

*• Using TMRE for the mitochondrial density estimates is a bit problematic because it brings mitochondrial membrane potential into the picture, whereas a mitochondrially targeted GFP or mitotracker green wouldn't have done so. I understand the need for a rapid, diffusible marker with the biopsies but the authors should discuss the limitations of this aspect of the study in the main body of the text.*

We agree that this is indeed a good point. In our original experimental design, we planned to use a dual injection of mitotracker green (MTG) and TMRE. However, the MTG dye, unlike TMRE, never diffused from the injection spot at room temperature, which made it impossible to use MTG for determining morphology. The reason why MTG is that sticky in human tissue is not yet clear, but it is possible that higher temperatures (37°C) are needed for the dye to spread uniformly. We had already briefly mentioned this in the original Materials and methods section, but have expanded this in the revised manuscript (subsection “Mitochondrial imaging”, third paragraph) and now also refer to this point in the Results section (subsection “Mitochondrial volume and numbers in submucous neurons do not differ between PD patients and controls”. Although we agree that the use of TMRE (at RT) for comparison of mitochondrial volume and numbers between PD patients and controls is not ideal, we believe that this pragmatic approach was justified based on the fact that there were no differences in TMRE signaling between the two populations.

The reviewer also mentions the possibility of expressing mitochondrially targeted GFP in the neurons. Unfortunately, this is technically not yet possible in our human enteric biopsies. The main problem is that the biopsies would need to be kept alive for much longer to allow a lentivirus- or AAV-based transduction approach to label mitochondria.

*• Was there any difference in mitochondrial morphology in the PD and control samples?*

As part of the analysis, we also measured ellipticity and sphericity of mitochondria. However, due to the limits of optical resolution, even after deconvolution, it was impossible to reliably determine the morphology of all mitochondria. Clusters of mitochondria were often identified as one, which made it impossible to interpret the values of ellipticity and sphericity. Therefore, we decided not to include these data in the manuscript. An example of this analysis is shown in Figure 7.

Author response image 1.Representative TMRE volume detection in the submucosal plexus.The first panel shows the typical volume detection (yellow) with additional spot detection (blue). The second picture is showing a color-coded image specific for sphericity. Within the white dotted line a cluster of mitochondria can be seen even if multiple spots were detected in this volume. Due to resolution limits, this cluster of mitochondria cannot be separated in individual mitochondria, not even after deconvolution. As seen from the color code, sphericity of this cluster is not representative for the sphericity of one mitochondrion. The third and fourth picture represent the same structure, representatively color-coded for ellipticity (oblate) and ellipticity (prolate).**DOI:**
http://dx.doi.org/10.7554/eLife.26850.027

*Reviewer #4:*

*Desmet et al. investigate enteric neuronal function from duodenal biopsies in patients with Parkinson's disease and controls. The goal is to explore whether functional assays can detect evidence consistent with the established increased prevalence of GI symptomatology and predilection for α-synuclein pathology in the enteric nervous system. The experiments do not detect any differences between cases and spouse controls, based on calcium imaging, mitochondrial membrane potential, and assessments of mitochondrial numbers. None of the biopsies had demonstrable α-synuclein aggregate pathology, making it difficult to conclude whether the negative results might be due to sampling issues. While innovative, the negative results in this small cohort, allow only preliminary conclusions to be drawn at this stage. This manuscript is not appropriate for eLife, and would be better suited to a neurology journal.*

Due to the nature of the work, which is at the interface of gastroenterology and neurology, and given the increasing interest in the gut (e.g. microbiome; inflammatory diseases) as a potential source or gateway for several diseases, we believe that this work would be of interest to a wider scientific audience than only clinical neurologists, and would like to defend our choice to submit to *eLife*.

*Suggestions for improving the manuscript:*

*It would be helpful to discuss further whether prior studies have found synuclein pathology with similar frequency in the submucosal vs. myenteric plexus.*

See our response to reviewers 1.

*Consider the possibility that pathology and dysfunction of the ENS may be "patchy", complicating interpretation of studies of selective biopsies.*

See our last response to reviewer 2.

*It would be interesting in the Discussion (or Introduction) to discuss any results from PD animal model studies of the GI tract.*

The Thy1-αSyn mouse, which overexpresses human wild-type α-synuclein, has been reported to have abnormal colonic motility (9), but otherwise we are not aware of any publications showing gastrointestinal abnormalities in the currently available genetic mouse models of PD.

Several studies have investigated the GI tract in toxin-based animal models of PD. Rats with a unilateral nigrostriatal lesion induced by intracerebral injection of 6-hydroxydopamine were reported to have a reduction in daily fecal output and a loss of nitrergic enteric neurons in the distal ileum and proximal colon, suggesting that loss of nigrostriatal neurons may induce secondary changes in the enteric nervous system (10). Chronic intragastric administration of low doses of rotenone was reported by Pan-Mantojo et al. to induce α-synuclein aggregates in the enteric nervous system of mice (11), whereas chronic oral administration of rotenone to mice did not induce α-synuclein aggregates but rather a decrease of α-synuclein expression in the enteric nervous system in the study by Tasselli et al. (12). MPTP, a compound that selectively kills nigrostriatal dopaminergic neurons, has been reported to also reduce the number of dopaminergic neurons in the myenteric and submucosal plexuses in mice and monkeys (13-15). However, data from these toxin-based models need to be interpreted with caution, because the relevance of these toxins for the pathogenesis of human PD is still questionable.

We have cited the animal work by Pan-Mantojo et al. in the manuscript, but would prefer not to add an extensive discussion of GI tract studies in PD animal models to the revised manuscript because our study focuses on human tissue and also because reviewers 3 and 4 suggested to shorten the Discussion.

*The discussion could be shortened.*

This was also suggested by reviewer 3. In the revised Discussion we have added several caveats for the interpretation of our data (the limitation of not having access to the myenteric plexus data, and the possibility of ‘patchiness’ of the pathology in the duodenum), as suggested by the reviewers. Nevertheless, we have managed to shorten the Discussion compared with the previous version (from a total of 1324 words to 1210 words).

*I suggest caution not to overstate the conclusion given the many caveats: "our findings… strong suggests that GI symptoms in PD do not arise from disturbed sub mucous neuronal function".*

We carefully screened the paper to tone down the interpretation of the results, mainly by removing words like “strongly” etc. We were careful not to overstate the results and further expanded the list of limitations of our study in the Discussion by adding a statement about the lack of myenteric data (third paragraph) and about the possibility of ‘patchiness’ and sampling errors (seventh paragraph).

*The authors also overstate the degree of controversy surrounding enteric nervous system pathology in PD (e.g. Introduction, second paragraph). While there is certainly debate about the potential utility as a clinical biomarker and the best staining protocol, I believe most experts agree that the ENS is pathologically involved.*

We agree that most researchers still believe that the ENS is pathologically involved in PD. As suggested by the reviewer, we have been careful not to overstate the degree of controversy surrounding enteric nervous system pathology in PD in the revised manuscript. In the revised Introduction we have replaced the phrase ‘Given the current controversy with respect to enteric α-synuclein immunohistochemistry…’ by ‘Given the current debate about the potential utility of enteric α-synuclein immunohistochemistry as a biomarker for PD…’. In the revised Discussion, we have replaced the sentence ‘So far, the outcome of these studies has been variable and controversial’ by ‘So far, the outcome of these studies has been variable’.

References:

1) Braak H, de Vos RA, Bohl J, Del Tredici K. Gastric α-synuclein immunoreactive inclusions in Meissner's and Auerbach's plexuses in cases staged for Parkinson's disease-related brain pathology. Neurosci Lett. 2006;396(1):67-72.

2) Lebouvier T, Chaumette T, Damier P, Coron E, Touchefeu Y, Vrignaud S, et al. Pathological lesions in colonic biopsies during Parkinson's disease. Gut. 2008;57(12):1741-3.

3) Lebouvier T, Neunlist M, Bruley des Varannes S, Coron E, Drouard A, N'Guyen JM, et al. Colonic biopsies to assess the neuropathology of Parkinson's disease and its relationship with symptoms. PLoS One. 2010;5(9):e12728.

4) Pouclet H, Lebouvier T, Coron E, des Varannes SB, Rouaud T, Roy M, et al. A comparison between rectal and colonic biopsies to detect Lewy pathology in Parkinson's disease. Neurobiol Dis. 2012;45(1):305-9.

5) Pouclet H, Lebouvier T, Coron E, Des Varannes SB, Neunlist M, Derkinderen P. A comparison between colonic submucosa and mucosa to detect Lewy pathology in Parkinson's disease. Neurogastroenterol Motil. 2012;24(4):e202-5.

6) Shannon KM, Keshavarzian A, Mutlu E, Dodiya HB, Daian D, Jaglin JA, et al. Α-synuclein in colonic submucosa in early untreated Parkinson's disease. Mov Disord. 2012;27(6):709-15.

7) Perry SW, Norman JP, Barbieri J, Brown EB, Gelbard HA. Mitochondrial membrane potential probes and the proton gradient: a practical usage guide. Biotechniques. 2011;50(2):98-115.

8) O'Reilly CM, Fogarty KE, Drummond RM, Tuft RA, Walsh JV. Quantitative analysis of spontaneous mitochondrial depolarizations. Biophys J. 2003;85(5):3350-7.

9) Wang L, Fleming SM, Chesselet MF, Taché Y. Abnormal colonic motility in mice overexpressing human wild-type α-synuclein. Neuroreport. 2008;19(8):873-6.

10) Blandini F, Balestra B, Levandis G, Cervio M, Greco R, Tassorelli C, Colucci M, Faniglione M, Bazzini E, Nappi G, Clavenzani P, Vigneri S, De Giorgio R, Tonini M. Functional and neurochemical changes of the gastrointestinal tract in a rodent model of Parkinson's disease. Neurosci Lett. 2009;467:203-7.

11) Pan-Montojo F, Anichtchik O, Dening Y, Knels L, Pursche S, Jung R, Jackson S, Gille G, Spillantini MG, Reichmann H, Funk RH. Progression of Parkinson's disease pathology is reproduced by intragastric administration of rotenone in mice. PLoS One. 2010;5(1):e8762.

12) Tasselli M, Chaumette T, Paillusson S, Monnet Y, Lafoux A, Huchet-Cadiou C, Aubert P, Hunot S, Derkinderen P, Neunlist M. Effects of oral administration of rotenone on gastrointestinal functions in mice. Neurogastroenterol Motil. 2013; 25:e183-e193.

13) Anderson G, Noorian AR, Taylor G, Anitha M, Bernhard D, Srinivasan S, Greene JG. Loss of enteric dopaminergic neurons and associated changes in colon motility in an MPTP mouse model of Parkinson’s disease. Exp Neurol. 2007;207:4-12.

14) Chaumette T, Lebouvier T, Aubert P, Lardeux B, Qin C, Li Q, Accary D, Bézard E, Bruley des Varannes S, Derkinderen P, Neunlist M. Neurochemical plasticity in the enteric nervous system of a primate animal model of experimental Parkinsonism. Neurogastroenterol Motil. 2009;21:215-22.

15) Natale G, Kastsiushenka O, Fulceri F, Ruggieri S, Paparelli A, Fornai F. MPTP-induced parkinsonism extends to a subclass of TH-positive neurons in the gut. Brain Res. 2010;1355:195-206.